# Detection of Coal Bust Risk in a Kilometer Depth Workface with Hard–Thick Roof Based on Active–Passive Seismic CT

Hu He [1,*], Junming Zhao [1], Ruyi Cheng [1], Zhengbing Men [1] and Zonglong Mu [2,3]

[1] School of Resources and Geosciences, China University of Mining and Technology, Xuzhou 221116, China
[2] School of Mines, China University of Mining and Technology, Xuzhou 221116, China
[3] Jiangsu Engineering Laboratory of Mine Earthquake Monitoring and Prevention, Xuzhou 221116, China
[*] Correspondence: hehu@cumt.edu.cn

**Abstract:** Practice and theory research proved the "square effect" during longwall mining with hard–thick strata lying on the coal seam, which could cause severe underground pressure and even dynamic disasters, such as coal burst; therefore, it became a key area and stage that need special attention. The combined active–passive seismic computed tomography (CT) was introduced to detect the abutment stress concentration in the coal seam. The results of active–passive CT inversion show that the "square effect" appears as early as the workface ahead of the theoretical position of 50 m with a 100 m significant influencing zone, which provides a strong guarantee for accurate risk evaluation of coal burst. Precursor information before the "square effect" can be identified based on the everyday total energy of mining tremors, which had period peaks in this case. The everyday average energy of mining tremors indicates that the seismic type during the "square effect" stage belonged to a foreshock–mainshock pattern. The combination of the seismic events and CT can improve the accuracy of coal burst danger distinguishing notably. The research can provide worthwhile guidance for the monitoring and prevention of coal burst hazards in similar conditions.

**Keywords:** hard–thick roof; square effect; coal burst; CT inversion; microseismic monitoring

## 1. Introduction

A coal burst, also called a rockburst, is a typical dynamic disaster during coal mining, with the distinctive features of sudden occurrence accompanied by strong vibration and destruction, which has become the most serious challenge of safety in deep mining [1]. Most mining countries, including China, Poland, the USA, Australia, and South Africa, are suffering from coal burst disasters [2–4]. According to statistics, there are approximately 138 burst-prone coal mines in China as of 2021, and their number keeps increasing every year [5]. Great achievements have been made on the mechanism, prediction, and control of this phenomenon, and dozens of theories have been proposed from different perspectives over the past decades [6–10]. A coal burst is a dynamic ejection of coal and/or surrounding rocks into the excavation. Sometimes it is accompanied by floor heaving or roof caving. Due to the cause of the occurrence, coal bursts are divided into a stress coal burst, stroke (roof) coal burst, and mixed stress–stroke coal burst. Some authors also distinguish a fault rock burst. [11–14]. Research shows that the majority of coal burst disasters are controlled and induced by the adjacent strong massive roof strata in China as the longwall mining method is used extensively [15–17]. Thus, the presence of strong massive roof strata close to the coal seam is taken as an indicator of coal burst hazard in practice and then, the analysis of the roof break law and monitoring of the stress changes during roof movement are of great significance for coal burst risk evaluation and prevention [18,19].

The monitoring of the precursor information before a coal burst is always a key issue and research hotspot since the first occurrence, and different kinds of devices and technologies have been developed and applied, of which micro-seismic monitoring, coal seam-stress monitoring, drilling bits, acoustic emission, electromagnetic emission are most

popular methods used in China coal mines [20,21]. Due to the complexity of coal burst, each technology has obvious pros and cons and can hardly describe the stress distribution in a wide area. Based on microseismic technology, seismic wave velocity computed tomography (CT) has been introduced and successfully used in coal mines and proved to be a powerful tool for the evaluation of dynamic hazards during underground mining [22–26]. Both "passive" and "active" velocity tomography have numerous reported studies and overcame lots of technical difficulties. Passive CT uses mining-induced tremors as the seismic sources and, therefore, can continuously and timeously carry out research with little limits but low resolution [27], while active CT is relatively time-consuming because of preparing artificial seismic sources, in most cases blasting but with high accuracy, and more importantly, active CT could be implemented before the longwall face excavation [28]. At present, for a specific longwall face, the passive and active CT were used independently, and few took both methods, that led us only to have partial features of stress concentration. In this paper, we combined the active and passive CT into one monitoring system to obtain stress information and assess the coal burst risk comprehensively in kilometer longwall face with a hard–thick roof so as to provide a basis for the design of prevention parameters and ensure the safety during the fracture and movement of the hard–thick roof.

## 2. Engineering Background

Zhangshuanglou Coal mine, located in the city of Xuzhou, Jiangsu Province in China, is a typical kilometer-deep mine that also encounters coal burst risk during excavation. The main factors influencing coal burst risk are coal seam depth, geological structure, as well as thick–hard roof. The longwall face LW23908 is the eighth working face in the No. 23 mining district, where the shallow area (south side) was finished and tuned into gobs; the coal pillars between coal faces are 5 m on average; a simplified schematic is shown in Figure 1. LW23908 exploits coal seam No. 9, which is 0.1~4.0 m thick in the whole mine, an average of 2.1 m, with a dip angle of 22~26°. The stratum in this panel is a monoclinic structure inclined towards NW, which is from the tail entry to the head entry in Figure 1. The size of the LW23908 is 190 m wide and 1180 m long, and the depth is over 1000 m, ranging from 1055 m to 1150 m. The immediate roof is gray–black mudstone with an average thickness of 1.0 m, and the main roof is a 40.8 m thick gray–white fine sandstone composed of quartz and feldspar. The features of the roof and floor strata are shown in Table 1. Both the coal seam and main roof have bursting liability tested based on the national standards of "Methods for test, monitoring, and prevention of rock burst–part 1: Classification and laboratory test method on bursting liability of roof strata, and part 2: Classification and laboratory test method on bursting liability of coal". The results are shown in Tables 2 and 3.

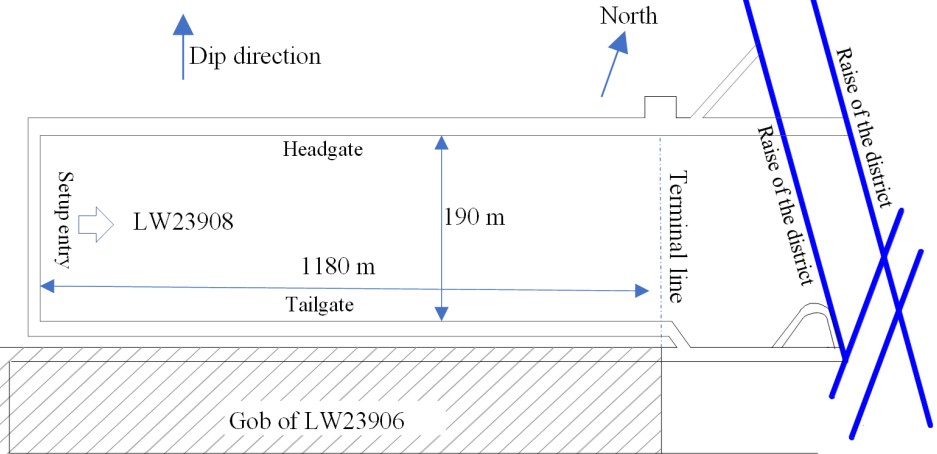

**Figure 1.** Sketch layout of LW23908.

**Table 1.** Features of roof and floor strata.

| Type of the Strata | Rock Category | Average Thickness/m | Lithological Characteristics |
|---|---|---|---|
| Main roof | Fine sandstone | 40.80 | Gray–white, relatively dense and hard, mainly composed of quartz and feldspar, with pyrite nodules and horizontal bedding. |
| Immediate roof | Mudstone | 1.0 | Gray–black, relatively dense, brittle, and fragile. |
| Immediate floor | Sandy mudstone | 10.55 | Dark gray, sandy argillaceous structure, uneven fracture surface with scratches and calcareous film, partial core fragmented. |
| Main floor | mudstone | 7.51 | Dark gray, dense and brittle, containing fragments of plant fossils. |

**Table 2.** Results of bursting liability of No. 9 coal seam.

| Index | | | | Testing Result | |
|---|---|---|---|---|---|
| Duration of Dynamic Fracture $DT$/ms | Elastic Strain Energy Index $W_{ET}$ | Bursting Energy Index $K_E$ | Uniaxial Compressive Strength $Rc$/MPa | Category | Level |
| 570 | 4.923 | 4.596 | 14.82 | II | Weak |

**Table 3.** Results of bursting liability of the roof strata.

| Index | | | | | | Testing Result | |
|---|---|---|---|---|---|---|---|
| Thickness $h$/m | Tensile Strength $R_c$/MPa | Modulus of Elasticity $E$/GPa | Uniaxial Compressive Strength $R_c$/MPa | Load on the Roof $q$/MPa | Bending Energy Index $U_{WQ}$/kJ | Category | Level |
| 44.7 | 5.86 | 9.70 | 51.84 | 1.3175 | 337.869 | III | Strong |

The coal burst risk degree of LW23908 is moderate based on the geological and mining technical conditions using the composite index method proposed by Prof. Liming Dou [29]. The dominant factor is the thick hard main roof which would cause severe dynamic disturbance to the coal seam during rupture and collapse, especially when the gob is a square, which is also named the "square effect" of the hard roof in some China coal mines. Historical data and experience of adjacent workfaces in the area also proved the "square effect" phenomenon; therefore, the mechanism, monitoring, and control of the "square effect" is the key to ensuring the safety of LW23908.

## 3. Detection of Coal Burst Risk Active–Passive Seismic CT

### 3.1. Active–Passive Combined CT Technology and System

As mentioned above, there is a positive correlation between the stress state and the seismic velocity, so we chose the seismic computed tomography (CT) technology to monitor the stress state in the coal seam during the process of roof "square effect". Passive tomography is using mining-induced seismic events as the source and can estimate the relatively high stress or rockburst hazard during the whole mining process based on seismic wave velocity inversion, while active tomography uses artificial seismic events such as blasting and hammer as the focus. The essence of active and passive CT is the inversion of seismic wave velocity, and for coal mines, P-wave velocity is always due to the high recognizability of the arrival time. Velocity CT depends on the relationship that seismic wave velocity along a ray is equal to the ray–path distance divided by the time to travel between the source and receiver, as shown in Figure 2. From this relationship, it is understood that time is the integral of the inverse velocity, 1/V, or slowness, S, as shown in Equation (1).

$$T_i = \int\limits_{Li} \frac{ds}{V(x,y)} = \int\limits_{Li} S(x,y)ds \tag{1}$$

where $T_i$ is the travel time; $L_i$ is the spread path of the *ith* seismic wave; $ds$ is the infinitesimal arc; $V$ is the velocity, and $S$ is the slowness. This equation is actually a typical nonlinear problem; if little changes in the velocity structure, the ray–path Li can be treated as a straight line; however, the path is usually a curve; in fact, due to the complexity of the rock mass, we need discrete the inversion area to N grids, so the travel time in the ith grid can be presented as Equation (2).

$$T_i = \sum_{j=1}^{N} a_{ij} s_j \tag{2}$$

where $a_{ij}$ is the length of the *ith* ray crossing the *jth* grid and $s_j$ is the slowness of each discretization cell. When a massive seismic ray path passes through the grid cells, arranging the travel time, distance, and slowness for each grid into matrices, the velocity can be determined through inverse theory, as shown in Equation (3).

$$\mathbf{AS = T} \tag{3}$$

$\mathbf{A} = (a_{ij})_{M \times N}$ is the distance matrix (M × N); $\mathbf{S}$ is the slowness matrix (M × N); $\mathbf{T}$ is the travel time per ray matrix (1 × N). Solving the equations, one can obtain the slowness distribution, thereby achieving the velocity structure inversion in the research zones.

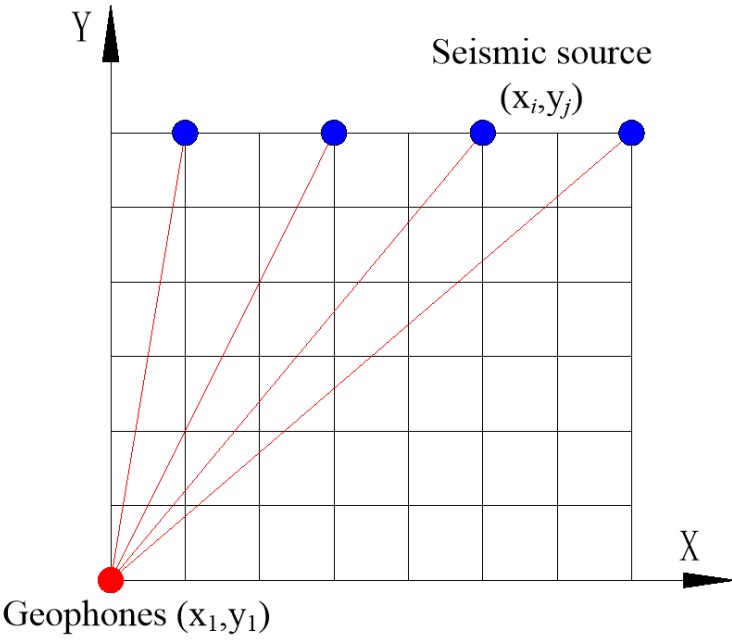

**Figure 2.** Schematic diagram of seismic velocity inversion.

Matrix inversion methods are effective but require considerable computational power for large datasets. Usually, the inverse problem is either underdetermined (more cells than rays) or overdetermined (more rays than cells) [30]. The most effective way to solve this problem is an iterative process. Currently, the most referenced iterative methods are Algebraic Reconstructive Technique (ART) [31] and Simultaneous Iterative Reconstructive Technique (SIRT) [32].

In order to combine the advantages of passive and active CT, equipment called KJ 470 was designed and developed, which could receive not only the artificial seismic sources, such as mechanical hammer and blasting, but also natural seismic events induced by mining activities, so the inversion accuracy and area are both boosted enormously. The KJ 470 system, normally 32 channels, is composed of the waveform acquisition and analysis system, monitoring substation, and geophones, all data is transmitted by optical fiber. The schematic diagram is shown in Figure 3. The frequency range of the geophones used in KJ 470 is 2 to 400 Hz. The geophones were installed on the bolts in the sidewall.

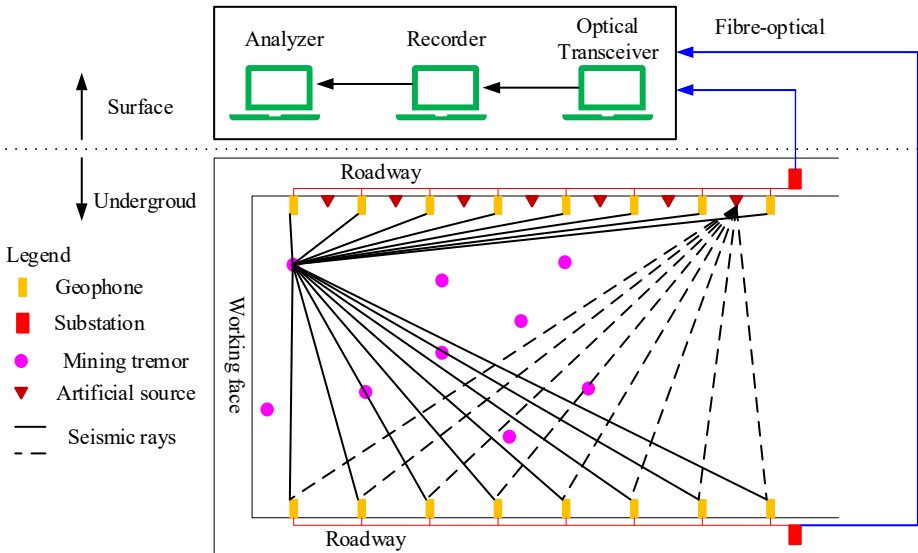

**Figure 3.** The schematic diagram of the dual-source CT system.

### 3.2. Implementation of Active–Passive CT

3.2.1. Active–Passive Seismic Wave CT Inversion at the Beginning Stage

According to the on-site conditions of LW23908, before the workface excavation, the geophone receivers and the artificial blasting points were all arranged in the roadways, a total of 22 geophones, of which 12 were installed in the headgate, and the others in the tailgate were used with an interval of 25 m, and two substations were placed in the chambers of the two entries, respectively. Blasting is designed as the active source in both entries, also with an equal interval of 25 m and 300 g explosive charges were used at each point. To increase the seismic ray's coverage area and inversion accuracy, the mining tremors were also introduced and selected for the inversion with energy greater than 1000 J, mostly in the range of $10^3$ to $10^5$ J, as shown in Figure 4. All the geophones will move forward as the mining advances to a certain distance, and the sources used for inversion are mainly mining tremors and a few actively blasting as auxiliary; therefore, we can get the distributions of the seismic wave during the excavation at different periods to understand the evolution of the stress state with a fairly high accuracy.

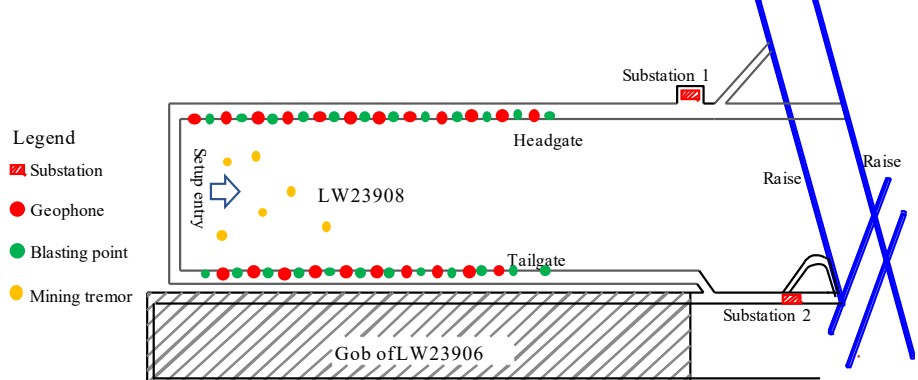

**Figure 4.** Schematic layout of active–passive detection.

As KJ 470 cannot calculate the energy of the tremors, so, first, the mining-induced tremors were located and calculated energy by the SOS microseismic system using the constant velocity model. When we obtained the basic information about the mining-induced tremors, we chose file directories of those with relatively high energy and clear P-wave arriving time. Then, in KJ470, we combined the blasting tremors and mining tremors to calculate the velocity and evaluate the reliability based on the first inversion;

the obtained velocities were used to calibrate the locations of the mining-induced tremors and analyze the errors, and the new locations were used as inversion sources; this process continued to loop, and ultimately, an optimal speed solution can be obtained. The location error of the mining tremors in LW23908 was about 10 m on average, based on the deep hole blasting verification.

From the time the mining begins, the first weighting of the main roof is called the beginning stage of the longwall face. The first active–passive CT detection was carried out on 18 March 2022, when the LW23908 had mined 24 m; a total of 24 blasting points and more than 70 mining tremors with energy exceeding 1000 J were used as the sources, no matter whether they were in the roof strata or in the coal seam. Figure 5 shows the ray grid generated by all the effective and reliable seismic events. The inversion grid was 10 m times 10 m. Finally, we obtained a 3D velocity distribution, but we only focused on the velocity in the coal seam, so we sliced the P-wave velocity in the coal seam to determine the high-stress areas, which were high rockburst areas correspondingly. Moreover, Figure 6 shows the slice of the P-wave velocity in the coal seam, and the high-velocity areas can be identified, which represent the relatively high-stress areas based on the positive correlation between each other [33–36]. In the beginning phase of the mining before the first weighting of the main roof, the high-stress area is located around the workface as the result of the stress redistribution and concentrated on the pillars. The envelope of the high seismic velocity is consistent with the theoretical analysis. The inversed velocity also indicates that the front and side abutment pressures of LW23908 are about 100 m and 30 m, respectively.

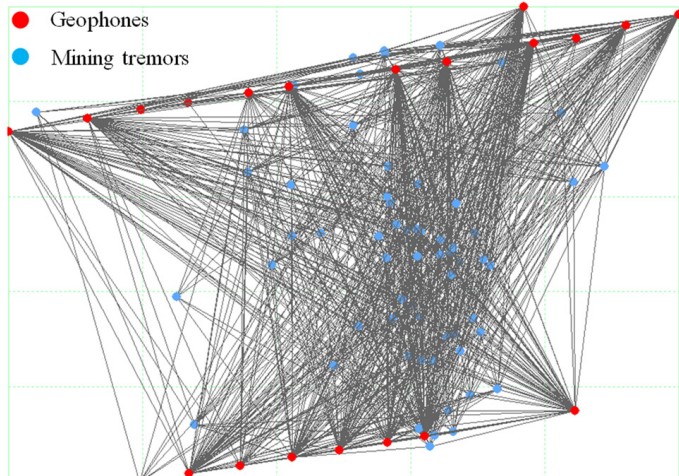

**Figure 5.** Seismic ray grid generated by both blasting and mining-induced tremors at the beginning stage of LW23908.

### 3.2.2. Active–Passive Seismic Wave CT Inversion at the "Square Effect" Area

On 10 May 2022, the LW23908 had mined forward 150 m and was approaching the "square effect" location, which is about 190 m from the setup entry by the theoretical analysis. Meanwhile, field practice has demonstrated that the influence of the "square effect" began about 50 m ahead square location, so the second active–passive CT detection was implemented. The data of mining tremors from 7 May 2022 to 10 May 2022 were selected as the focus, and after pretreatment, a total of 183 tremors with clear P-wave arrival were picked, accompanied by artificial blasting sources. The density of the active–passive seismic rays can ensure the accuracy requirements. Figure 7 shows the isoline of the inversed P-wave velocity. Two high P-wave velocity areas are identified, which demonstrate two high-stress zones in the coal seam. It is not difficult to see that these two high-stress zones (high-stress area 1 and high-stress area 2) correspond to the rear and front foundations of the hard–thick roof, which indicates that the hard roof is about to rupture and cave soon, the deformation and subsidence of the roof strata causing the stress

concentration in the coal seam. The distribution of the P-wave velocity is highly consistent with the theoretical analysis of the "square effect" and verified the existence of the "square effect" again. For LW23908, the influencing area of the "square effect" is about 150 m, and the stress concentration degree in front of the workface is higher than that behind the workface. Compared with the inversion results on 18 March 2022, shown in Figure 6, the maximum of the P-wave increases notably from 4.7 km/s to 5.5 km/s, indicating a much more dangerous coal burst during the "square effect" phase than the beginning stage; the everyday total energy and numbers of mining-induced tremors of LW23908 also proved that the "square effect" stage is the biggest challenge of coal burst prevention.

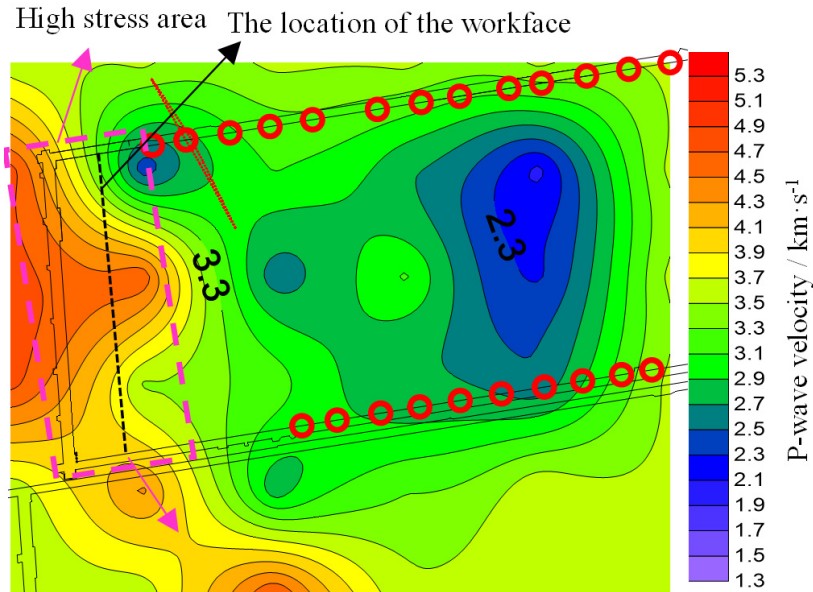

**Figure 6.** Cloud map of the P-wave velocity in the coal seam at the beginning stage of LW23908.

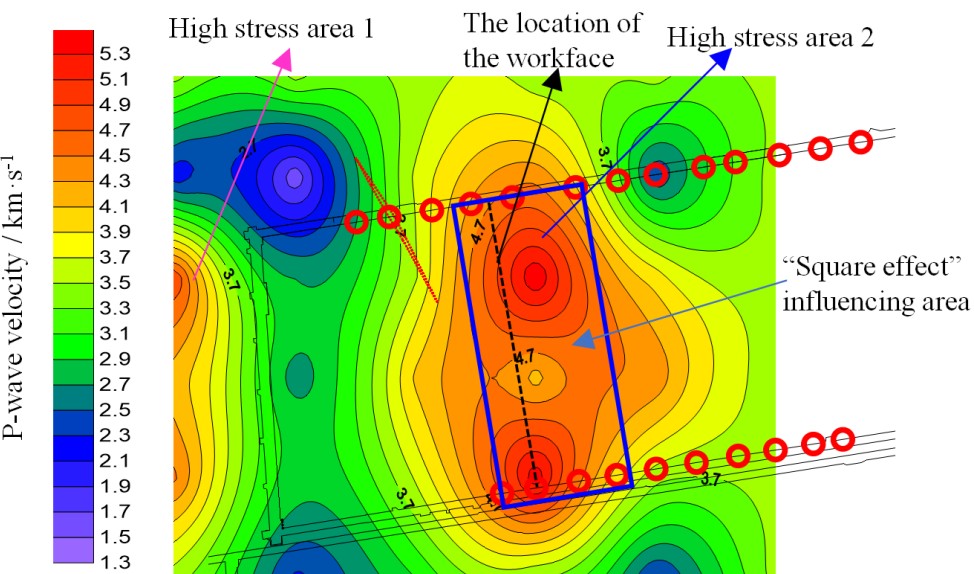

**Figure 7.** Cloud map of passive wave CT inversion on May 10.

### 3.3. Coal Burst Prevention Aiming the "Square Effect" Based on the CT Inversion

Based on the P-wave velocity distribution inversed by active–passive CT, the high-stress areas, which stand for the highly risky area of coal burst, can be determined during the "square effect" stage, as shown in Figure 7, and destress scheme can be designed more accurately. As analyzed above, the main reasons for the high stress and coal burst

danger are the depth and hard–thick roof; therefore, the hard–thick roof fracturing is the most feasible solution, and coal seam large-diameter destress is the auxiliary. Aiming the high-stress area 2 detected on 10/5/2022, the technical measures and parameters were as follows:

(a)     Coal seam large-diameter borehole destress. The destress boreholes with a diameter of 150 mm were 25 m deep into the coal seam, a spacing interval of 2.0 m, and carried out in both roadways;

(b)     Deep-hole blasting of hard–thick roof. Three blasting holes as a group, marked as 1#, 2#, and 3#, distributing a fan shape on the section, as shown in Figure 8. The 1# and 3# blasting holes are both 35 m long with an elevation of 5° and 75°, respectively, while 2# is 50 m with a 45° elevation. The diameters of the three blasting holes are all 89 mm; the charge length of 1#, 2#, and 3# blasting holes are 10 m, 15 m, and 10 m that are 40 kg, 60 kg, and 40 kg explosives, and the diameter of the explosive tube is 65 mm. The interval between the blasting group is 10 m; therefore, a total of 20 groups were implemented in each roadway. The induced cracks developed after blasting, which can reduce the strength of the hard–thick roof and the coal burst danger effectively [37]. The deep-hole blasting used to fracture the hard-thick roof can stimulate obvious tremors in the roof strata and could be recorded by the SOS microseismic system, but the energy of each blasting varies significantly, which reflects both the stress state and the difference in blasting effect. In this case, tremors provoked by deep-hole blasting can reach $10^5$ J and most in the range of $10^3 \sim 10^5$ J.

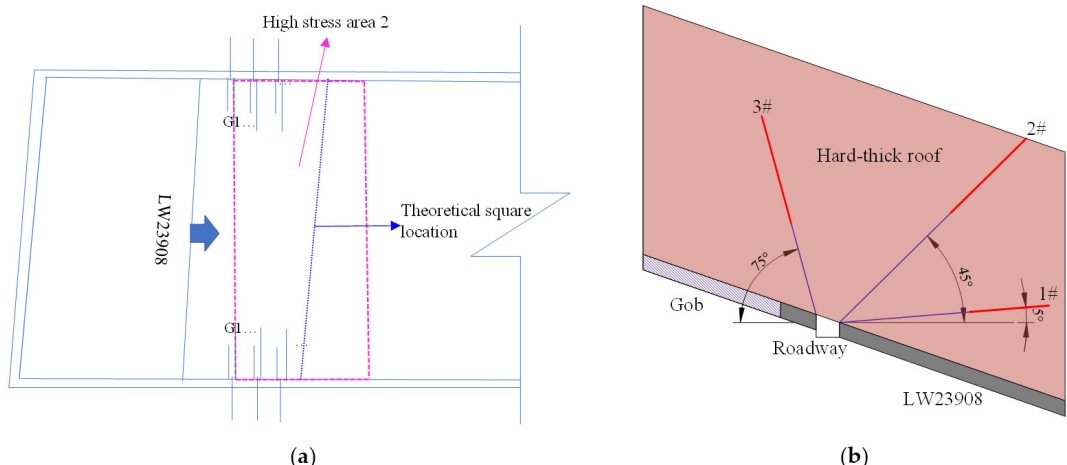

(**a**)                                                              (**b**)

**Figure 8.** Design of the deep-hole blasting of the hard–thick roof. (**a**) Plane sketch of deep-hole blasting program. (**b**) Profile layout of deep-hole blasting borehole.

## 4. Discussion

### 4.1. Mechanism of the "Square effect" of Thick–Hard Roof

In longwall mining, the thick–hard roof usually has a large caving interval according to the key strata key theory, and when the weighting interval equals the length of the working face, it means that the gob is a square at this time, the underground pressure reaches its maximum, causing severe ground pressure behavior even coal burst sometimes, called the "square effect" [38]. Actually, the "square effect" is the result of the breaking and movement of a thick–hard roof. We can use the plate model to explain this phenomenon, as shown in Figure 2. It is known that most of the overlying coal seams are sedimentary strata that are characterized by obvious layers and weak planes, which separate easily under shearing stress; therefore, even the thick–hard roof can be regarded as a thin plate to analyze the stress distribution under different boundary conditions, as shown in Figure 9,

and we can obtain the maximum normal stress both on the strike (*x*-axis,) and (*y*-axis) dip direction, as Equation (4).

$$
\begin{cases}
\sigma_{x\max} = \dfrac{12\mu q a^4 b^2}{\pi^2 h^2 \left(3a^4 + 2a^2 b^2 + 3b^4\right)}, \; x = 0, a; y = b/2 \\[2mm]
\sigma_{y\max} = \dfrac{12\mu q a^2 b^4}{\pi^2 h^2 \left(3a^4 + 2a^2 b^2 + 3b^4\right)}, \; x = a/2; y = 0, b
\end{cases}
\tag{4}
$$

where $\mu$ is the Poisson's ratio; $q$ is the overlying load on the roof; $a$ and $b$ are the overhang length and width of the roof, can be simplified as the advancing distance and width of the working face, respectively, and $h$ is the thickness of the roof plate.

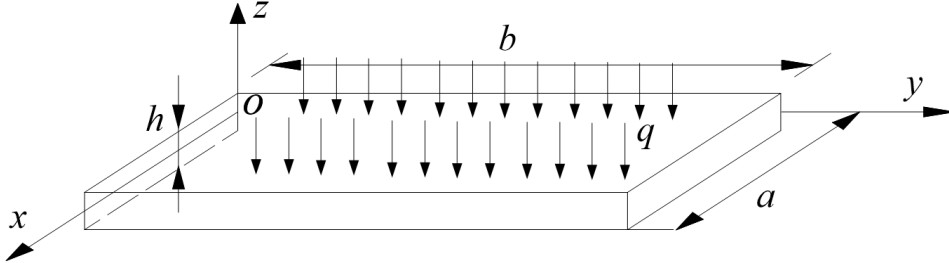

**Figure 9.** Elastic thin plate model of the hard roof.

Based on Equation (3), at the early stage of mining excavation, that is $a \leq b$, $\sigma_{x\max}$ is always greater than $\sigma_{y\max}$; therefore, $\sigma_{x\max}$ plays a vital role in the breaking of the roof. Although the thickness of the thick–hard roof can be tens of meters, the fracture and cave of the roof under longwall mining is usually in layers, and statistics showed that when the thickness of a single layer is more than 5 m, it can be referred to as hard-collapsed roof and would form a large area of a suspended roof, so we assumed the plate thickness h = 5 m in the simplified model. Assuming $\mu$= 0.25, *h*= 5.0 m, we can obtain a trend graph of $\sigma_{x\max}/q$ with the advancing of mining in different widths of working face, as shown in Figures 10 and 11. Figure 10 shows that the normal stress always reaches the maximum stress at the midpoint of the long side first under uniform load and has a positive correlation with the length of the working face. Figure 11 shows the relation of the $\sigma_{x\max}/q$ with the advancing distance of that $\sigma_{x\max}/q$ increased with the advancing distance rapidly to its maximum when $a = b$, which means that the roof plate was a square now, and at this moment, the roof was most likely to break and cave causing much more violent ground pressure behavior than usual, that was figuratively called the "square effect" by the mining engineers in China. Equation (4) and Figure 11 explain this phenomenon by a simplified mechanical model.

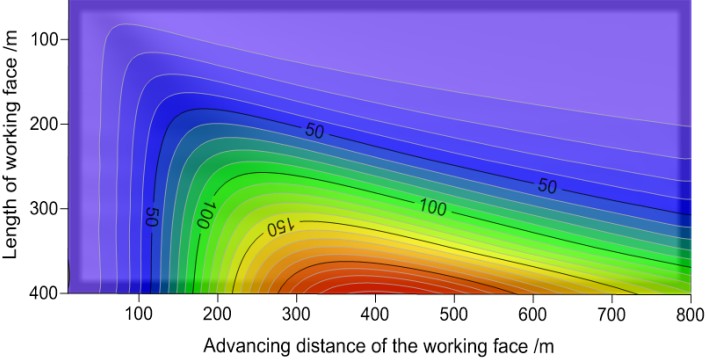

**Figure 10.** Normal stress of *x*-axis with the advancing of mining in different widths of working face.

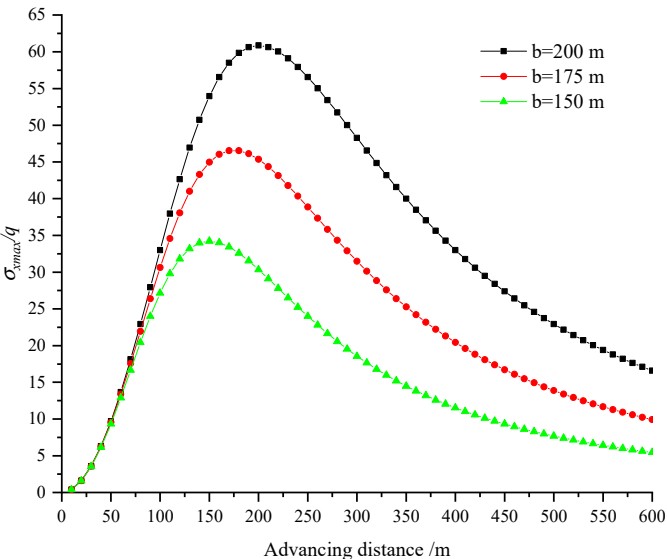

**Figure 11.** Maximum normal stress of *x*-axis with the advancing of mining with different widths of working face.

### 4.2. The Evolution of Seismic Events during the "Square Effect" Period

The 32-channel micro-seismic system SOS developed by the Central mining institute of Poland was installed at the Zhangshuanglou coal mine and used to record the mining tremors; thereby, the coal burst risk can be evaluated based on the seismicity data [39–41]. The CT inversion revealed that the "square effect" began when LW23908 advanced 150 m on 10 May 2022, ahead of the theoretical square location of 40 m; immediately after the CT inversion, the destress program was completed in the high-stress area 2. On 26 May 2022, the workface advanced to 190 m, equivalent to the length of the workface, where the goaf formed a standard square, and the underground pressure should have increased sharply. Figure 12 presents the total energy and frequency of seismic events per day from 5 May to 6 June. Distinct periodic peaks on 15 March 2022, 20 May 2022, and 24 May 2022 appeared before the "square effect", which indicated the rapid energy release of the surrounding strata and was mainly caused by the deformation and rupture of the hard–thick roof. The microseismic monitoring results are able to provide useful precursor information before the "square effect" and, combined with the CT inversion, give a much more accurate early warning for the period and scope of the coal burst risk. After the destress projects, the total everyday energy that is less than 105 J before and at the "square effect" is reduced significantly compared with other similar workfaces. Of course, the advanced judgment of the high-stress area and reasonable design of destress program have played a crucial role. When LW23908 passed the theoretical location of the "square effect" on 26 May, the influence still sustained for four days until 30 May and then fell to a normal level. Figure 13 shows the everyday average and maximum energy of the mining tremors from 10 May to 13 June 13, a different trend which can be found easily with the evolution of total energy before the theoretical "square effect" location that no obvious precursor patterns can be identified and hardly predict the coal burst risk. However, on 26 May, the average energy reached the maximum value for this period of time, although the total energy on this day was the lowest, which indicated that the seismic type at LW23908 belonged to a foreshock–mainshock pattern with no aftershock. As Figure 13 shows, the average energy of everyday mining tremors in these three stages is mainly less than 1500 J. Meanwhile, the maximum energy of no more than 1000 J, which suggests an extremely low coal burst risk, on the other hand, also proves the effectiveness of the destress.

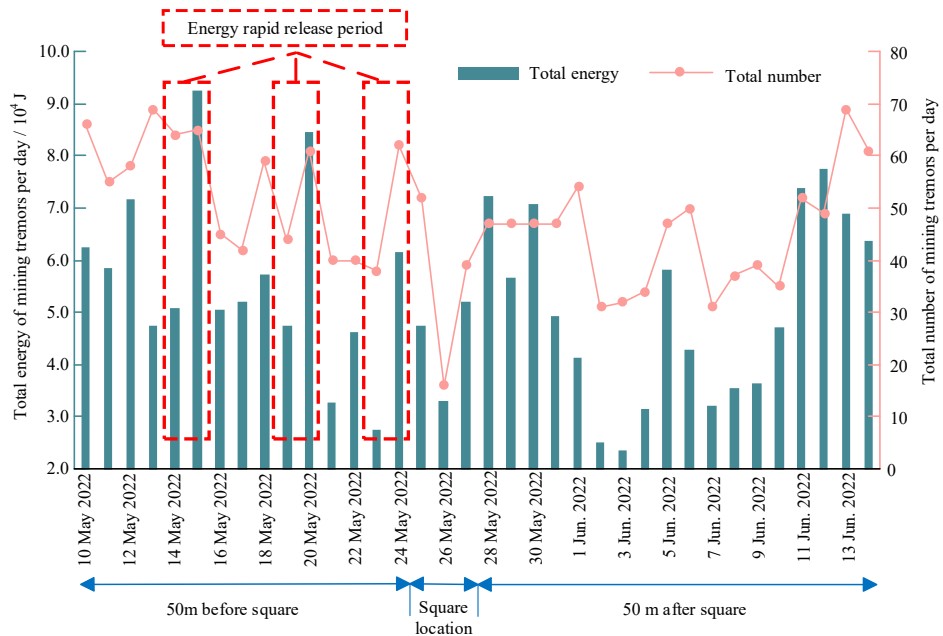

**Figure 12.** Total energy and frequency of mining tremors before and after "square effect".

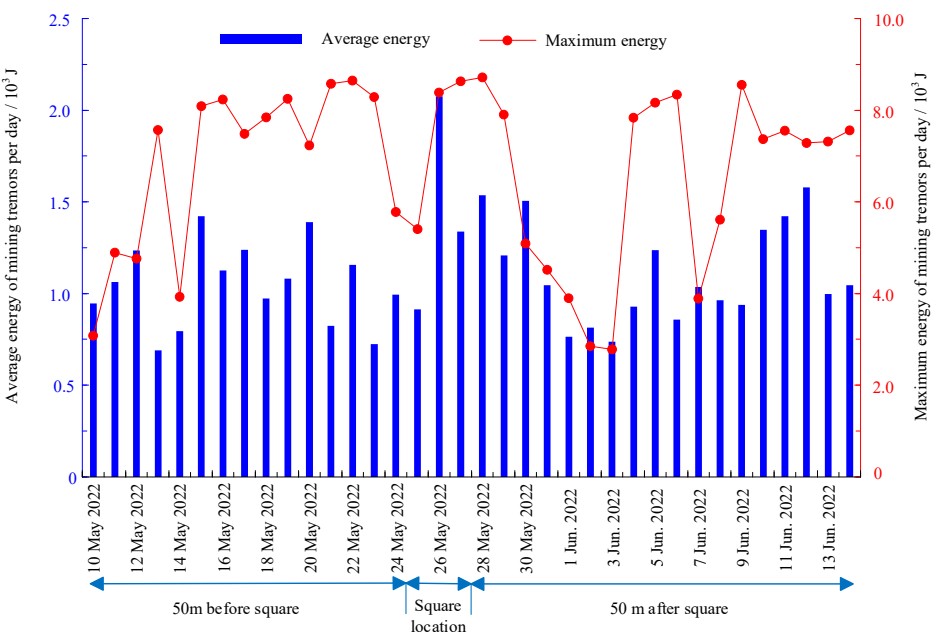

**Figure 13.** Average and maximum energy of mining tremors before and after "square effect".

## 5. Conclusions

(1) The "square effect" exists in longwall coal mining workfaces that have hard–thick main roofs near the coal seam; a simplified plate model was established to explain this phenomenon, which indicates that when the workface length equals the advancing distance, where the planar shape of the gob forms a square, the normal stress of the plate reaches its maximum and is prone to rupture and cave, causing violent strata behaviors;

(2) The joint active–passive seismic CT technology and equipment were introduced at a typical kilometer coal burst longwall face LW23908 with hard–thick main roof to investigate the stress distribution in the coal seam at different periods during the mining. The inversed P-wave velocity indicates that at the beginning stage of mining, the front and side abutment pressures of LW23908 are about 100 m and 30 m,

respectively, and the "square effect" appears as early as the workface ahead of the theoretical position at 50 m with a 100 m significant influencing zone, which provides a strong guarantee for accurate risk evaluation of coal burst;

(3) Based on the detection of the high-stress areas, supplementary targeted destress projects can be designed, and for LW23908 coal seam, large-diameter borehole destress and deep-hole blasting of hard–thick roof were carried out immediately in the high-stress area. The microseismic system was used to monitor and verify the effect of destress by analyzing the total energy, average energy, and maximum energy as well as frequency of mining tremors, precursor information before the "square effect" could be identified based on total energy features, which had period peaks in this case. The total everyday energy was always less than 105 J before and at the "square effect," which has proved the effectiveness of the destress program;

(4) The average energy indicates that the seismic type during the "square effect" stage at LW23908 belonged to a foreshock–mainshock pattern; therefore, the precursor information can be used to provide an early warning of coal burst, and active–passive combined seismic CT technology can delineate the scope of the high stress. The combination of these two, seismic events and CT, can improve the accuracy of coal burst danger, distinguishing notably;

(5) The fracturing could change the break and movement of the hard–thick roof, overall reducing the cave intensity and impact on the coal seam; after fracturing, the energy resale mode during the break is changed from centralized to decentralized, manifested as massive low-energy microseismic events dominating even at the "square effect" stage, thus avoiding the accumulation of excessive elastic energy and releasing in an extremely short time.

**Author Contributions:** Conceptualization and methodology, H.H.; investigation, J.Z. and R.C.; data curation, Z.M. (Zhengbing Men); software, Z.M. (Zonglong Mu). All authors have read and agreed to the published version of the manuscript.

**Funding:** This research was funded by the State Key Research Development Program of China (Grant No. 2022YFC3004605), the National Natural Science Foundation of China (No. 51974302; 52274147), and the Fundamental Research Funds for the Central Universities (No. 2013QNB30).

**Institutional Review Board Statement:** Not applicable.

**Informed Consent Statement:** Not applicable.

**Data Availability Statement:** The data used for conducting classifications are available from the corresponding author upon request.

**Acknowledgments:** The authors would like to thank the engineers Lei Zhang, Wuxian Luo, Bing Li, and Yuanwei Cao from Xukuang Energy Co., Ltd. (Xuzhou, Jiangsu Province, China) for their valuable contributions to the paper.

**Conflicts of Interest:** The authors declare no conflict of interest.

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
