# Peer review of "Detection of Coal Bust Risk in a Kilometer Depth Workface with Hard–Thick Roof Based on Active–Passive Seismic CT"

_applsci, doi:10.3390/app13106173_

Round 1

Reviewer 1 Report

Line 2-3: What does "kilometer workface" mean? Does it mean the length of the longwall panel or the depth of excavation? The authors later in the text (lines 71-72) write that: “the LW23908 is (…) 1180 m long and the depth is over 1000 m, ranges from 1055 m to 1150 m”. Please explain. Maybe the title should be modified because it is not clear? Maybe it's better to write that it's about an excavation at a great depth or something similar?

Line 33-35: The authors write “Although there is still no consensus on the mechanical model and mechanism of coal burst…”. I think this sentence needs to be modified. Coal burst is a dynamic ejection of coal and/or surrounding rocks into the excavation. Sometimes it is accompanied with floor heave or roof caving. Due to the cause of occurrence, rock bursts are divided into stress rockburst, stroke (roof) rockbursts and mixed stress-stroke rockbursts. Some authors also distinguish a fault rock burst.

Line 70: Did the thickness of the coal seam No. 9 change from 0.1 to 4 m within the selected longwall panel or was it the whole mine? This is a very wide range. Please explain.

Line 71: The dip angle of the coal seam is quite significant, i.e. 22-26 degrees. In which geographical direction did the coal seam dip in the longwall area? Please explain.

Line 86: Please provide references for the composite index method.

Line 111: What does sj mean in formula (2)? Is it slowness in cell j? Please add a description in the text.

Line 116: Is S the slowness matrix? Please add a description in the text.

Line 121: The authors use the term voxel used in three-dimensional graphics. Is the velocity distribution reconstructed in 2D or 3D? Please explain. Maybe the word cell would be better.

Lines 123-124: Please provide references for ART and SIRT.

Line 125: The authors did not explain what passive seismic tomography is all about. Authors should add a few sentences of explanation about passive seismic tomography with references.

Line 127: What load of explosives was used during each blasting? Please explain.

Line 138: Please write what frequencies are registered by the geophones used. Are they installed in the sidewall, roof or floor of the opening? Please explain.

Lines 136-146: In order to locate the tremors' foci, it is necessary to assume the propagation velocity of seismic waves in the rock mass. Was the propagation velocity adopted ad hoc or was it due to active seismic tomography? Please explain. What was the error in the location of the tremors' foci in the area of the longwall panel LW23908? Please clarify.

Line 152: What were the energies (magnitudes) of the induced tremors?

Lines 152-153: What cell sizes were used to reconstruct the velocity distribution? Moreover, did the use of tremors as a source of seismic waves result from the close proximity of the thick sandstone to the coal seam? The tremors could have originated in the seam, but also in the sandstone above the seam. If the seam-sandstone vertical distance was large, then foci may be high above the seam. Please explain.

Line 162. The authors write that the ray grid came from both blasting and mining-induced tremors, while in Fig. 5 only tremors are marked. Please mark the position of blasts in Fig. 5 and add a description.

Line 165: The velocity of a P-wave in a coal seam at a depth of 1 km is about 2.1-2.2 km/s. In Fig. 6, this velocity equals even 5 km/s, which corresponds to compact and strong rocks such as sandstones. What can cause such a high P-wave velocity in coal seam, significantly exceeding the corresponding critical stress. Were rock bursts recorded during mining of the longwall panel LW23908?

Line 172: What were the energies (magnitudes) of the induced tremors?

Lines 184-185: The obtained P-wave velocities correspond to compact and strong rocks. Does the presented velocity distribution apply to the sandstone deposited in the roof of seam No. 9? Were foci of induced tremors in this sandstone? How does this relate to the considered tremors caused by blasting in the coal seam? Please explain.

Lines 209-210: Please describe the energies (magnitudes) of the tremors provoked.

Line 233: Why was the plate thickness h=5 m assumed? Please explain.

Lines 275-278: Please describe whether strong tremors, i.e. with energies greater than or equal to 1E5 J, were recorded during longwall mining? The average values are of course informative, but from the point of view of the risk of rock bursts, the energy of the strongest tremors is important. Did strong tremors occur before the use of rock burst prevention, and after its application they were no longer present? Please explain.

Author Response

Dear Reviewer,

Thank you for the valuable comments and suggestions on our manuscript. We significantly checked and revised the manuscript according your comments. In addition, detailed revisions are listed below point by point.

Your sincerely,

Hu He

School of Resources and Geosciences

China University of Mining & Technology

Xuzhou 221116, Jiangsu, China

Response to Reviewer 1 Comments

Point 1: Line 2-3: What does "kilometer workface" mean? Does it mean the length of the longwall panel or the depth of excavation? The authors later in the text (lines 71-72) write that: “the LW23908 is (…) 1180 m long and the depth is over 1000 m, ranges from 1055 m to 1150 m”. Please explain. Maybe the title should be modified because it is not clear? Maybe it's better to write that it's about an excavation at a great depth or something similar?

Response 1: Thank you very much for the suggestion. Yes, the “kilometer” in the title means the depth of the workface. It is much more clear using “kilometer depth workface”, so we plan to revise the title as “Detection of Coal Bust Risk in a Kilometer depth with Hard-Thick Roof based on Active-Passive Seismic CT”.

Point 2: Line 33-35: The authors write “Although there is still no consensus on the mechanical model and mechanism of coal burst…”. I think this sentence needs to be modified. Coal burst is a dynamic ejection of coal and/or surrounding rocks into the excavation. Sometimes it is accompanied with floor heave or roof caving. Due to the cause of occurrence, rock bursts are divided into stress rockburst, stroke (roof) rockbursts and mixed stress-stroke rockbursts. Some authors also distinguish a fault rock burst.

Response 2: Much appreciated for the excellent revision, we’ll adopt this modification.

Point 3: Line 70: Did the thickness of the coal seam No. 9 change from 0.1 to 4 m within the selected longwall panel or was it the whole mine? This is a very wide range. Please explain.

Response 3: The thickness change of No. 9 coal seam in in the whole mine as the data we quote usually from the geologic report of the whole mine and hardly got the accurate data before the workface excavation.

Point 4: Line 71: The dip angle of the coal seam is quite significant, i.e. 22-26 degrees. In which geographical direction did the coal seam dip in the longwall area? Please explain.

Response 4: The strata in this panel is a monoclinic structure inclined towards NW, which is from the tailentry to the headentry in Figure 1, and we’ll add a compass in Figure 1 of the revised version.

Point 5: Please provide references for the composite index method.

Response 5: The composite index method for rockburst danger evolution was proposed in the Ph.D. thesis “Modyfikacja klasyfikacji stanow zagrozania tapaniami w kopalniach wegla” in 1998, Universiyt Slesia, which was written in Polish, and we can hardly find the original thesis, while the Chinese version is much more common in rockburst relevant books, so we have provided a Chinese book as the reference. (Dou, L., Mu, Z., Cao, A., Gong, S., He, H., Lu, C. Rockburst prevention and control of coal mine: Science Press, Beijing. China, 2017;168-173.)

Point 6: Line 111: What does sj mean in formula (2)? Is it slowness in cell j? Please add a description in the text.

Response 6: Yes, sj means the slowness of each discretization voxel. We’ll add a description in the text.

Point 7: Line 116: Is S the slowness matrix? Please add a description in the text.

Response 7: Yes, S means the slowness matrix. Many thanks for the kind reminder.

Point 8: Line 121: The authors use the term voxel used in three-dimensional graphics. Is the velocity distribution reconstructed in 2D or 3D? Please explain. Maybe the word cell would be better.

Response 8: The model and velocity inversion were in 3D, but in this case we did not used the velocity in the overly strata because of ray density too sparese and only the distribution in coal seam were analyzed. So, actually we only obtained the 2D velocity and the “cell” is more accurate.

Point 9: Lines 123-124: Please provide references for ART and SIRT.

Response 9: The following references have been added to the article, (1) Gilbert, P. Iterative methods for the three-dimensional reconstruction of an object from projections. J Theor Biol. 1972, 36,0105-117. (2) Dines, K., Lytle, R. Computerized geophysical tomography. Proceedings of the IEEE, 1979, 67(7), 1065-1076.

Point 10: Line 125: The authors did not explain what passive seismic tomography is all about. Authors should add a few sentences of explanation about passive seismic tomography with references.

Response 10: Thanks for the suggestion. The description “Passive tomography is using mining induced seismic events as the sources and can estimate the relatively high stress or rock burst hazard during the whole mining process based on seismic wave velocity inversion” is going to added in the text.  

Point 11: Line 127: What load of explosives was used during each blasting? Please explain.

Response 11: 300 g explosive charges were used at each point.

Point 12: Line 138: Please write what frequencies are registered by the geophones used. Are they installed in the sidewall, roof or floor of the opening? Please explain.

Response 12: The frequency range of the geophones used in KJ470 is 2-400 Hz. The geophones were installed on the bolts in the sidewall.

Point 13: Lines 136-146: In order to locate the tremors' foci, it is necessary to assume the propagation velocity of seismic waves in the rock mass. Was the propagation velocity adopted ad hoc or was it due to active seismic tomography? Please explain. What was the error in the location of the tremors' foci in the area of the longwall panel LW23908? Please clarify.

Response 13: Yes, that’s a key point to ensure the accuracy. As KJ470 cannot calculate the energy of the tremors, so first, the mining induced tremors were located and calculated energy by the SOS microseismic system using the constant velocity model. When we obtained the basic information of the mining induced tremors, we chose file directories of those with relative high energy and clear P-wave arrive time. And then in KJ470 we combined the blasting tremors and mining tremors together to calculate the velocity and evaluate the reliability, based on the first inversion, the obtained velocities were used to calibrate the locations of the mining induced tremors again and analyze the errors, and the new locations were used inversion sources, this process continues to loop, ultimately an optimal speed solution can be obtained. The location error of the mining tremors in LW23908 was about 10 m on average based on the deep hole blasting verification.

Point 14: Line 152: What were the energies (magnitudes) of the induced tremors?

Response 14: The mining induced tremors selected for the inversion were greater than 1000 J, mostly in the range of 103 to 105 J.

Point 15: Lines 152-153: What cell sizes were used to reconstruct the velocity distribution? Moreover, did the use of tremors as a source of seismic waves result from the close proximity of the thick sandstone to the coal seam? The tremors could have originated in the seam, but also in the sandstone above the seam. If the seam-sandstone vertical distance was large, then foci may be high above the seam. Please explain.

Response 15: The inversion grid was 10 m times 10 m. Actually, the tremors we used both from the coal and the roof/floor strata, and yes, a lot of them were located in the sandstone above the coal seam.

Point 16: Line 162. The authors write that the ray grid came from both blasting and mining-induced tremors, while in Fig. 5 only tremors are marked. Please mark the position of blasts in Fig. 5 and add a description.

Response 16: The blasting points were in the tailgate and headgate as shown in Figure 4, so normally we only display the rays of the induced tremors to check if the ray density satisfied the accuracy requirements.

Point 17: Line 165: The velocity of a P-wave in a coal seam at a depth of 1 km is about 2.1-2.2 km/s. In Fig. 6, this velocity equals even 5 km/s, which corresponds to compact and strong rocks such as sandstone. What can cause such a high P-wave velocity in coal seam, significantly exceeding the corresponding critical stress. Were rock bursts recorded during mining of the longwall panel LW23908?

Response 17: The high P-wave velocity can reflect the stress state, which has a positive correlation between each other. According to the inversion results, there were indeed some very high P-wave velocity zones, almost equaled to that in the hard sandstone, and we analyzed these zones indicated high risk of rockburst. Once the high risk was detected, some destress methods must be implemented and even mining production must be stopped, so during the mining of LW23908 no rockbursts occurred.

Point 18: Line 172: What were the energies (magnitudes) of the induced tremors?

Response 18: Similar to Point 14, the mining induced tremors selected for the inversion were greater than 1000 J, mostly in the range of 103 to 105 J.

Point 19: Lines 184-185: The obtained P-wave velocities correspond to compact and strong rocks. Does the presented velocity distribution apply to the sandstone deposited in the roof of seam No. 9? Were foci of induced tremors in this sandstone? How does this relate to the considered tremors caused by blasting in the coal seam? Please explain.

Response 19: As we mentioned above, the model and velocity inversion were in 3D, mining tremors with energy greater than 1000 J in the LW23908 mining area were used as the seismic sources, no matter they were in the roof strata or in the coal seam. And finally we obtained a 3D velocity distribution, but we only focused on the velocity in the coal seam, so we sliced the P-wave velocity in the coal seam and then to determine the high stress areas which were high rockburst areas correspondingly.

Point 20: Lines 209-210: Please describe the energies (magnitudes) of the tremors provoked.

Response 20: The deep-hole blasting used to fracture the hard thick can stimulate obvious tremors in the roof strata and could be recorded by the SOS microseismic system, but the energy of each blasting varies significantly, which reflects both the stress state and the difference of blasting effect. In this case, tremors provoked by deep-hole blasting can reach 105 J and most in the range of 103 ~105 J.

Point 21: Line 233: Why was the plate thickness h=5 m assumed? Please explain.

Response 21: Although the thickness of the thick hard roof can be tens of meters, the fracture and cave of the roof under longwall mining were usually in layers, and statistics showed that when the thickness of a single layer more than 5 m, it can be referred as hard-collapsed roof and would form a large area of suspended roof, so we assumed that plate thickness h=5 m in the simplified model.

Point 22: Lines 275-278: Please describe whether strong tremors, i.e. with energies greater than or equal to 1E5 J, were recorded during longwall mining? The average values are of course informative, but from the point of view of the risk of rock bursts, the energy of the strongest tremors is important. Did strong tremors occur before the use of rock burst prevention, and after its application they were no longer present? Please explain.

Response 22: That’s pretty important information, in fact no rockbusrt occurred during the mining of LW23908. As we all know that the safety managements are really strict now in China’s coal mine, most time in order to ensure the absolute safety, excessive destress projects were conducted and implemented from the mining beginning to the end, so neither before nor after the application of rockburst prevention has there been any rockburst event.

Reviewer 2 Report

Overall this is an interesting paper, with valuable field measurements.  During my 40 years in coal mining, I have often heard "the square effect" mentioned, but this is the first paper to address it directly.  However, the data presented in the paper does not entirely convince me of the authors' conclusions.  Figure 12 shows the key data, but it only shows the results for 100 m of longwall panel extraction.  I think data from a much longer period would be better.  Also, the fact that the roof was de-stressed at this time means that we can't be sure a decrease in the seismic energy release was due to passing the square, or just the de-stressing.  

On figures 6 and 7, please show where the longwall face is located.  I also think you should point out more clearly that the seismic velocities shown in figures 6 and 7 are in the roof, not the coal seam.

Overall the English is pretty good.  Only one glaring error--in line 186 I think you mean "beginning" where you say "begging."  Otherwise the English was not exactly standard but I could follow it without too much trouble.

Author Response

Dear Reviewer,

Thank you for the valuable comments and suggestions on our manuscript. We significantly checked and revised the manuscript according your comments. In addition, detailed revisions are listed below point by point.

Your sincerely,

Hu He

School of Resources and Geosciences

China University of Mining & Technology

Xuzhou 221116, Jiangsu, China

Response to Reviewer 2 Comments

Point 1: Figure 12 shows the key data, but it only shows the results for 100 m of longwall panel extraction. I think data from a much longer period would be better. Also, the fact that the roof was de-stressed at this time means that we can't be sure a decrease in the seismic energy release was due to passing the square, or just the de-stressing.

Response 1: Thank you very much for the suggestion. Yes, it is really a pity that we cannot prove the “sqaure effect” clearly and directly for various reasons especially related to safety. As we all know that the safety managements are really strict now in China’s coal mine, most time in order to ensure the absolute safety, excessive destress projects were conducted and implemented from the mining beginning to the end, some of the rules of the roof movements have indeed been covered up.

Point 2: On figures 6 and 7, please show where the longwall face is located.  I also think you should point out more clearly that the seismic velocities shown in figures 6 and 7 are in the roof, not the coal seam.

Response 2: Thank you very much for the suggestion.We will add the location of the longwall face in Figure 6 and 7. The model and velocity inversion were in 3D, mining tremors with energy greater than 1000 J in the LW23908 mining area were used as the seismic sources, no matter they were in the roof strata or in the coal seam, and blastings were all in the coal seam. And finally we obtained a 3D velocity distribution, but we only focused on the velocity in the coal seam, so we sliced the P-wave velocity in the coal seam and then to determine the high stress areas which were high rockburst areas correspondingly. In fact, we sliced of velocity in different levers, but as you mentioned, it is hard to say the velocity shown in Figure 6 was completely the P-wave velocity of the coal seam, it should be a reflection of a comprehensive influences to the coal seam. This is indeed an important topic to study deeply for us in future especially under theses complex geological conditions.

Reviewer 3 Report

Dear Authors,

1) Correct the numerical sequence of the equations;

2) Engineering background: Is it "Materials and Methods"?

Author Response

Dear Reviewer,

Thank you for the valuable comments and suggestions on our manuscript. We significantly checked and revised the manuscript according your comments. In addition, detailed revisions are listed below point by point.

Your sincerely,

Hu He

School of Resources and Geosciences

China University of Mining & Technology

Xuzhou 221116, Jiangsu, China

Response to Reviewer 3 Comments

Point 1: Correct the numerical sequence of the equations.

Response 1: Thank you very much for the suggestion. We’ll comprehensively check and correct the numerical sequence of the equations.

Point 2: Engineering background: Is it "Materials and Methods"?

Response 2: Yes, this part is equivalents to the “"Materials and Methods", but from traditional habits and consistency with the content we still refer to it as “Engineering background”.

Round 2

Reviewer 1 Report

Dear Authors, 

thank you for considering my comments. In my opinion, the article has been improved, and it is more understandable now. The obtained results may be of practical use.

Still, the very high p-wave velocity (even for a highly stressed coal seam) is interesting. Perhaps this is due to the reconstruction technique itself. 

Sincerely